# Pressure-controlled magnetism in 2D molecular layers

Yulong Huang [1] ✉, Arjun K. Pathak [2] ✉, Jeng-Yuan Tsai[3], Clayton Rumsey [4], Mathew Ivill[5], Noah Kramer[2], Yong Hu[1], Martin Trebbin [4,6], Qimin Yan [3] ✉ & Shenqiang Ren [1,3,4,6] ✉

Long-range magnetic ordering of two-dimensional crystals can be sensitive to interlayer coupling, enabling the effective control of interlayer magnetism towards voltage switching, spin filtering and transistor applications. With the discovery of two-dimensional atomically thin magnets, a good platform provides us to manipulate interlayer magnetism for the control of magnetic orders. However, a less-known family of two-dimensional magnets possesses a bottom-up assembled molecular lattice and metal-to-ligand intermolecular contacts, which lead to a combination of large magnetic anisotropy and spin-delocalization. Here, we report the pressure-controlled interlayer magnetic coupling of molecular layered compounds via chromium-pyrazine coordination. Room-temperature long-range magnetic ordering exhibits pressure tuning with a coercivity coefficient up to 4 kOe/GPa, while pressure-controlled interlayer magnetism also presents a strong dependence on alkali metal stoichiometry and composition. Two-dimensional molecular interlayers provide a pathway towards pressure-controlled peculiar magnetism through charge redistribution and structural transformation.

Two-dimensional (2D) magnetic materials[1-4], are ubiquitous in a range of spintronic applications from data storage and quantum sensing to spin electronics[5-9]. Stimuli-responsive atomically thin magnets are key to device miniaturization because of the favorable tuning capabilities and profoundly distinct material properties from their bulk counterparts[10-13]. Such discovery has ignited the search for high-performing 2D molecular magnetic alternatives resembling classical magnets through a bottom-up molecular assembly approach[14]. Magnetic anisotropy and magnetization are governed by molecular building blocks originating from its structure and symmetry, as well as intramolecular and intermolecular bond interactions including bond length and geometries, etc[15,16]. However, the control of magnetic textures and anti-symmetric exchange interactions were not observed in molecular layered magnets, while it is challenging to manipulate through the synthetic chemistry approach. One potential promising pathway to harness magnetic anisotropy and magnetization is by controlling relatively weak interlayer exchange interactions as compared to intralayer through pressure effect[17-21].

## Results

Here we report the pressure tuning magnetic anisotropy and magnetization of 2D layered metal-organic magnet $Li_{0.7}Cr(pyz)_2Cl_{0.7}$ (LCPC, pyz = pyrazine)[22,23] and its potassium substitution $K_xCr(pyz)_2Cl_x$ (KCPC) compound, which exhibit magnetic ordering temperature $T_c$ of 510 K and 480 K, respectively at ambient pressure and magnetic field of 5 kOe. Room-temperature magnetic ordering of such 2D molecule-based magnets arises from the strong $d\text{-}\pi^*$ direct spin interactions of Cr-pyrazine layers, as a result of the antiparallel spins of Cr and N from

[1]Department of Mechanical and Aerospace Engineering, University at Buffalo, The State University of New York, Buffalo, NY 14260, USA. [2]Department of Physics, SUNY Buffalo State, Buffalo, New York 14222, USA. [3]Department of Physics, Northeastern University, Boston, MA 02115, USA. [4]Department of Chemistry, University at Buffalo, The State University of New York, Buffalo, NY 14260, USA. [5]DEVCOM Army Research Laboratory, Aberdeen Proving Ground, MD 21005, USA. [6]Research and Education in Energy, Environment and Water (RENEW) Institute, University at Buffalo, The State University of New York, Buffalo, NY 14260, USA. ✉e-mail: yhuang59@buffalo.edu; pathakak@buffalostate.edu; q.yan@northeastern.edu; shenren@buffalo.edu

pyrazine[24–27]. The remnant magnetic moment of Cr in LCPC is estimated as 1.34 $\mu_B$ at 5 K[22]. Nitrogen atoms from pyrazine molecules contribute the antiparallel spin against that of chromium. With the introduction of hydrostatic pressure, magnetic interaction increases due to the enhancement of interlayer coupling from the reduction of interlayered lattice (Fig. 1a) between magnetic Cr-pyrazine layers. A representative transmission electron microscopy (TEM) image of KCPC magnet reveals such 2D layered structure (Fig. 1b). We further compare the lattice dimension effect between KCPC and LCPC because potassium possesses a larger ionic radius (152 pm) than that of lithium cation (90 pm). Indeed, a pronounced pressure effect on the magnetic exchange interaction is observed in KCPC, where the coercivity coefficient under pressure is defined as the pressure derivative of magnetic coercivity ($dH_c/dP$). The KCPC magnet shows a larger coercivity coefficient of over 4 kOe/GPa at room temperature than that of -1 kOe/GPa in LCPC (Fig. 1c). The coercivity coefficient increases with temperature and shows the positive pressure effect for the measured temperature range from 5 K to 375 K, presenting a at least double pressure effect compared to other magnets[28–31].

The structural formation and 2D magnetism of LCPC and KCPC compounds result from the redox reaction between an alkali metal and the precursor $Cr(pyz)_2Cl_2$[22,23]. With the formation of LCPC layered compounds, the bonding breaks between chromium and chlorine ions in $Cr(pyz)_2Cl_2$, while inducing a new layer of LiCl (Fig. 2a). Such

structural transformation from space group Immm[24] to P4/mmm[22] is indicated by time-dependent X-ray diffraction (XRD) patterns (Fig. S1). During the redox reaction, the ultraviolet-visible absorption spectra are captured when the $Cr(pyz)_2Cl_2$ precursor is coordinated with lithium cation (Fig. 2a). The absorption peak at 550 nm after 44 minutes suggests the lithiation which is absent in the precursor, while accompanying with alkali-metal stoichiometry-dependent phase formation from a magnetically soft to hard phase, as shown in Fig. 2b. At 140 minutes, a homogenous hard phase is obtained with the coercivity of -3.7 kOe. The soft magnetic behavior is captured in the in situ magnetic measurement (Fig. S2), considering as an initial state of alkali-metal diffusion. A large coercivity is observed at room temperature by the reduction of precursor $Cr(pyz)_2Cl_2$ by alkali metals. The incident room-temperature magnetism indicates the new phase due to alkali-metal lithiation process.

The 2D layered structure is the peculiar feature of LCPC and KCPC magnets. The scanning electron microscopic (SEM) images (Fig. 3a, b and Figs. S3a–f), indicate the layered sheets stacking together and 2D feature in the LCPC and KCPC magnets. The expanded layered structure is obtained when alkali metals reduce Cr cation in $Cr(pyz)_2Cl_2$ precursor, leaving a large tunable space among interlayers. The alkali-metal diffusion in Cr-pyz network is confirmed by energy-dispersive spectra (Fig. S3g, h). The TEM images (Fig. S4) and selected area electron diffraction (SAED) patterns of LCPC and KCPC sheets (the

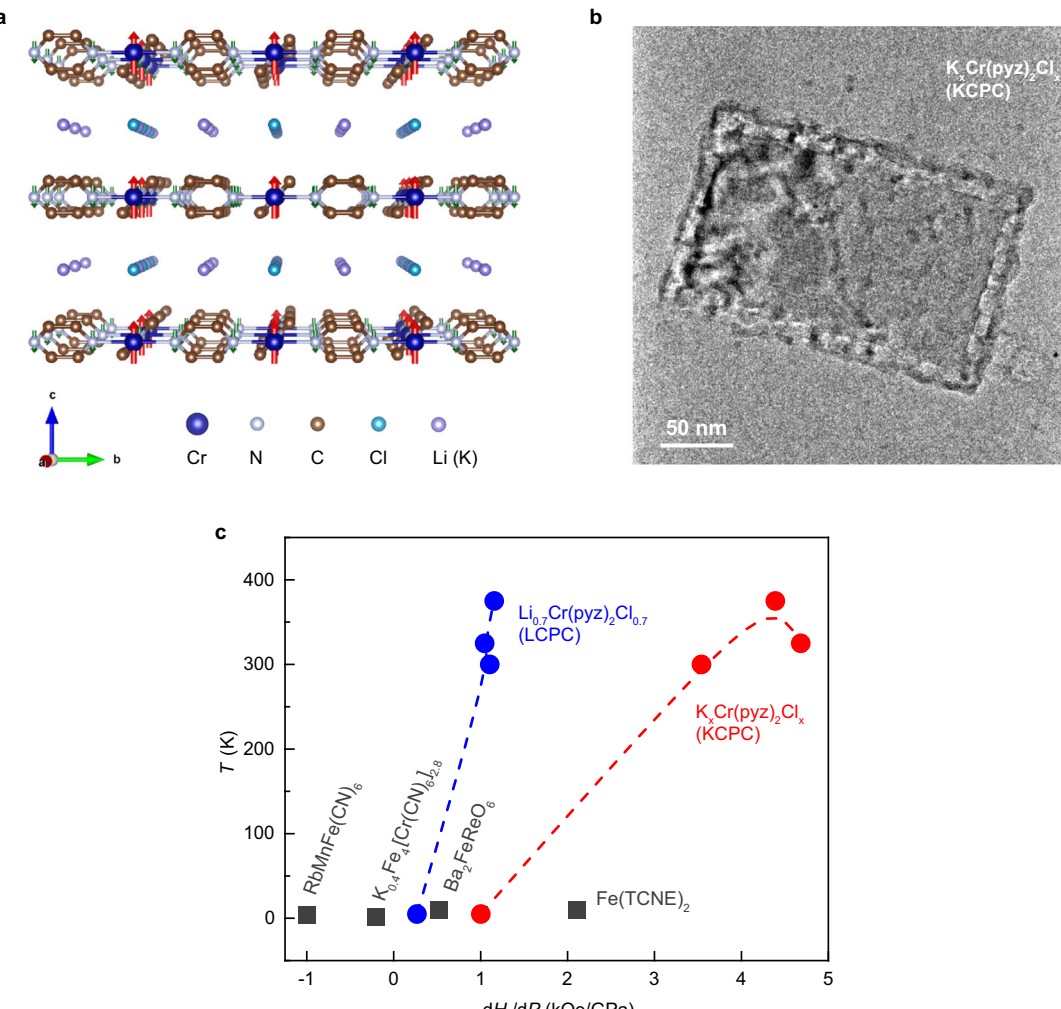

**Fig. 1 | Pressure effect on metal-organic magnets LCPC and KCPC. a** The crystal structure of LCPC or KCPC shows the layered structure. **b** A representative TEM image of KCPC magnet indicates a 2D structure. **c** Comparison of the coercivity coefficient of inorganic and organic magnets presents the large tunability in LCPC and KCPC magnets under pressure. The coercivity coefficient is defined as the pressure derivative of the magnetic coercivity ($dH_c/dP$).

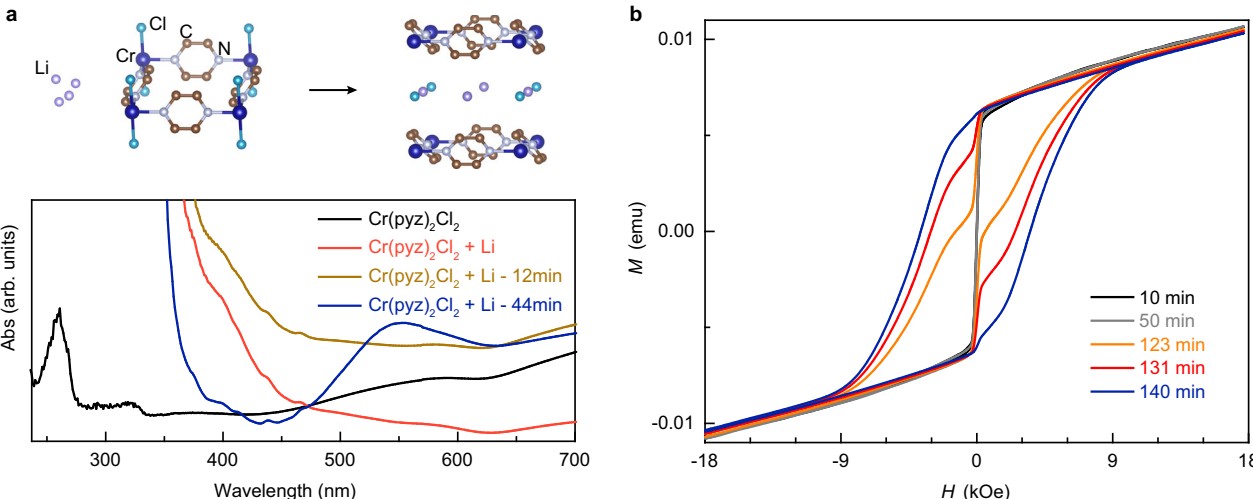

**Fig. 2 | Structural characterization of LCPC magnet. a** The structural transformation is illustrated by the procedure of the reaction between lithium and precursor Cr(pyz)$_2$Cl$_2$. The UV-Vis spectra of the LCPC reaction solution indicate the absorption development with time. **b** The in situ $M$-$H$ loops of LCPC magnet were measured at room temperature from the beginning of the solution reaction.

**Fig. 3 | The in situ process of the formation of magnetism and structural coordination.** SEM images and SAED patterns show the morphologies and crystallinity of **a** LCPC and **b** KCPC magnets. **c** Raman spectra show the shift of molecular vibration peaks between LCPC and KCPC magnets. The solid red and black lines are the smoothed results. **d** FTIR spectra of LCPC and KCPC magnets indicate the absorption of molecular vibration changes with alkali-metal substitution.

insets in Fig. 3a, b) confirm their high crystallinity and in-plane 2D structure. The LCPC and KCPC magnets both possess a tetragonal structure with a space group of P4/mmm[22] which is represented by XRD patterns (Fig. S5). The substitution of lithium by potassium leads to a larger interlayer spacing. Raman spectra in Fig. 3c show the four strong internal vibration modes of pyrazine molecule in LCPC and KCPC magnets, which can be assigned to the in-plane bending of the pyrazine ring ($\delta_{ring}$ ~650 cm$^{-1}$), the carbon-hydrogen bond ($\delta_{CH}$ around 1200 cm$^{-1}$), and the stretching of pyrazine ring ($v_{ring}$ around 1000 cm$^{-1}$ and $v'_{ring}$ ~1600 cm$^{-1}$). The vibration modes that appear below 600 cm$^{-1}$ are assigned to Cr-N (Fig. S6). These assignments are based on Raman scattering data of isolated pyrazine molecule[32] and metal-pyrazine complexes of similar frameworks[33]. It should be noted that the $\delta_{ring}$ and $v_{ring}$ modes in LCPC shift to lower frequencies compared to those in KCPC; while the $\delta_{CH}$ and $v'_{ring}$ modes present an opposite shift. Such difference in Raman data of pyrazine molecules also occurs in metal-pyrazine complexes where the one of a high-spin state shows a lower Raman vibration frequency compared to the one of a low-spin state[33,34]. The frequency shift in LCPC and KCPC could be ascribed to charge redistribution on pyrazine molecules due to the modified interlayer magnetic interaction. The alkali-metal-induced changes in electronic structure are also shown in Fourier-transform infrared (FTIR) spectra. Figure 3d presents the FTIR spectra of LCPC and KCPC magnets at room temperature which share a common adsorption peak at 995.2 cm$^{-1}$. The LCPC magnet shows the vibrational modes of tetrahydrofuran ligand[35–37] at 1051.1 cm$^{-1}$ and 890.8 cm$^{-1}$, which shift to 1058.5 cm$^{-1}$ and get diminished in KCPC magnet, respectively. The same peak shift is also observed in the adsorption mode at 760.4 cm$^{-1}$ in LCPC magnet and 767.8 cm$^{-1}$ in KCPC magnet. Besides, thermal effect on KCPC magnet (Fig. S7) is weaker than that in LCPC as reported in our previous study[23]. Therefore, alkali-metal substitution modifies not only crystal lattice, but also the feature of electronic structure.

The ionic radii and composition of alkali-metal cations in the ordered lattice dictate magnetic anisotropy and magnetization of molecular layered magnets. Figure 4a shows the magnetic field-dependent magnetization of LCPC and KCPC magnets at room temperature, where the LCPC exhibits a larger coercivity and a higher magnetization. In addition, the LCPC shows a higher $T_c$ than that of KCPC magnet (Fig. 4b), resulting from a large spacing from the potassium cations. With decreasing temperature, both coercivity and magnetization are enhanced in KCPC magnet (Figs. S8, 9), while the coercivity can reach 12.6 kOe at 10 K. First-order reversal curves (FORCs) and derived diagrams reveal inhomogeneity, switching field distribution and coercivity distribution of magnetic materials[38]. Figure 4c, d show the FORCs of LCPC and KCPC magnets, respectively, which are measured by increasing external magnetic field $H$ from a reverse field $H_r$ to the maximum field of 18 kOe at room temperature. From the major magnetization curve measured in a largest magnetic field range, the coercivity of LCPC magnet is 8.9 kOe at ambient pressure, which is larger than that of KCPC magnet. The magnetization $M$ on each FORC is varied by $H$ and $H_r$, therefore the FORC distribution function $\rho$ is a second-order derivative of magnetization, $\rho(H, H_r) = -\frac{\partial}{\partial H_r}(\frac{\partial M}{\partial H})$. As exhibited in Fig. 4e, f, the function $\rho$ of LCPC and KCPC magnets are plotted with the axes of $H$ and $H_r$. In the FORC diagram of LCPC magnet (Fig. 4e), a remarkable and narrow spot appears around $(H, H_r) = (9.5 \text{ kOe}, -9.5 \text{ kOe})$, corresponding to the magnetization curves of large slopes. The grey line of $H_r = -H$ crosses over the narrow FORC spot of LCPC magnet, implying the coercivity $H_c$ can be reached when the applied $H_r$ has the same magnitude of $H_c$. The non-fully symmetric FORC spot of LCPC magnet indicates that the appearance of $H_c$ demands a much larger $H_r$. In the FORC diagram of KCPC magnet (Fig. 4f), an equal $H_r$ can allow the appearance of $H_c$ since the spot is perfectly symmetric with respect to the line of $H_r = -H$. The FORC spot region in KCPC magnet is longer and closer to the origin

point than that in LCPC magnet, revealing a smaller coercivity and less homogeneity of magnetic domains in the former.

Figure 5a shows the hydrostatic pressure effect on layered molecular magnets to control magnetic interaction at both intra- and interlayer. In the Cr-pyrazine layer, the reduced lattice along a- and b-axes modifies magnetic interaction bridged by pyrazine molecules. Meanwhile, the shortened dimension along the c-axis enhances interlayer magnetic coupling. Magnetic hysteresis (M-H) loops of LCPC magnet present enhanced magnetism under hydrostatic pressures at 375 K (Fig. 5b), as well as at lower temperatures (Fig. S10). The coercivity $H_c$ of LCPC magnet at 375 K increases from 4.3 kOe to 5.5 kOe with an increase of pressure from 0.200 GPa to 1.242 GPa. The magnetization of LCPC magnet is increased up to 31.6 emu/g (375 K) and 45.1 emu/g (5 K) at 50 kOe under 1.242 GPa (Fig. S10). The enhanced magnetic coercivity and magnetization in the LCPC magnet result from the spin coupling of the Cr-pyrazine interlayers. The temperature-dependent magnetic susceptibility of KCPC magnet under hydrostatic pressure reveals a large enhancement in magnetism with the pressure increasing from 0.147 GPa to 0.664 GPa. This enhancement is reversible as indicated by the reduced magnetic susceptibility of the KCPC magnet after decreasing pressure to 0.427 GPa (Fig. 5c). Similar pressure enhancement is also observed in KCPC magnets (Fig. S11–13). The coercivity $H_c$ of KCPC magnet develops with the hydrostatic pressure and temperature, as shown in Fig. 5d. Instead of a linear $H_c$-$P$ dependence in LCPC magnet for all temperatures, KCPC magnet shows a large and nonlinear coercivity increase with hydrostatic pressure. An anomaly appears in KCPC magnet at 0.664 GPa where the $H_c$ largely increases for 325 K and 375 K. Such enhancement in coercivity and magnetization is almost absent in the precursor Cr(pyz)$_2$Cl$_2$ (Fig. S14), which is magnetically ordered below 55 K[24]. The negligible pressure effect in the precursor Cr(pyz)$_2$Cl$_2$ is unexpected since its 2D monolayer is calculated for an enhanced magnetic ordering temperature of 350 K[39].

The pressure results in repeatable jumps in magnetization of layered KCPC magnet (Fig. 6 and Fig. S11–13). The hydrostatic pressure of 0.147 GPa induces the sharp jumps in magnetization at $H_1 = 23$ Oe and $H_2 = 931$ Oe in the M-H loop of KCPC magnet (Fig. 6a) that are not present in the ambient pressure. At these magnetic fields, magnetization flip occurs rapidly in KCPC magnet. With the hydrostatic pressure increasing to 0.488 GPa, $H_1$ maintains while $H_2$ shifts to 28 Oe. Such sharp magnetization flip can be regarded as the Barkhausen jump[40,41], which could be caused by the rapid changes of the size and orientation of magnetic domain. Similar jumps in single-layer SrRuO$_3$ can be manipulated by He-ion irradiation[41]. With increasing the hydrostatic pressure to 0.664 GPa, the Barkhausen jump is absent in KCPC magnet at 375 K; meanwhile, a classic and smooth M-H loop appears with $H_c$ of 2.1 kOe. This smooth M-H loop maintains up to 0.752 GPa with a small kink at zero magnetic field. The jump in magnetization indicates that the magnetic domain is tuned by hydrostatic pressure at the same thermodynamic scale. The temperature-dependent M-H loops reveal the thermodynamic jump in magnetization under a constant pressure of 0.664 GPa in KCPC magnet (Fig. 6b). With the decrease of the temperature to 325 K, the jump in magnetization appears with $H_1 = -0.5$ kOe, while exhibiting a smaller coercivity $H_c$ of 1.2 kOe at 325 K than that of 2.0 kOe at 375 K. At a lower temperature of 300 K, the jump in magnetization exists at multiple magnetic fields, $H_1 = -0.6$ kOe, $-1.9$ kOe, $-2.6$ kOe and $H_2 = 1.8$ kOe, while the KCPC magnet shows a smooth M-H loop at 5 K with $H_c$ of 10.9 kOe. The jump in magnetization exists in a range of temperature and pressure, suggesting a metamagnetic behavior. Wide-angle X-ray scattering (WAXS) patterns reveal the sharper and stronger peaks in the KCPC after pressure (Fig. 6c and S15), indicating a higher crystallinity due to the pressure effect. The obtained peaks (012), (013), (231), and (133) in KCPC after pressure are shifting to a higher scattering vector $q$ compared to those of the pristine sample (the inset in Fig. 6c), revealing a

 

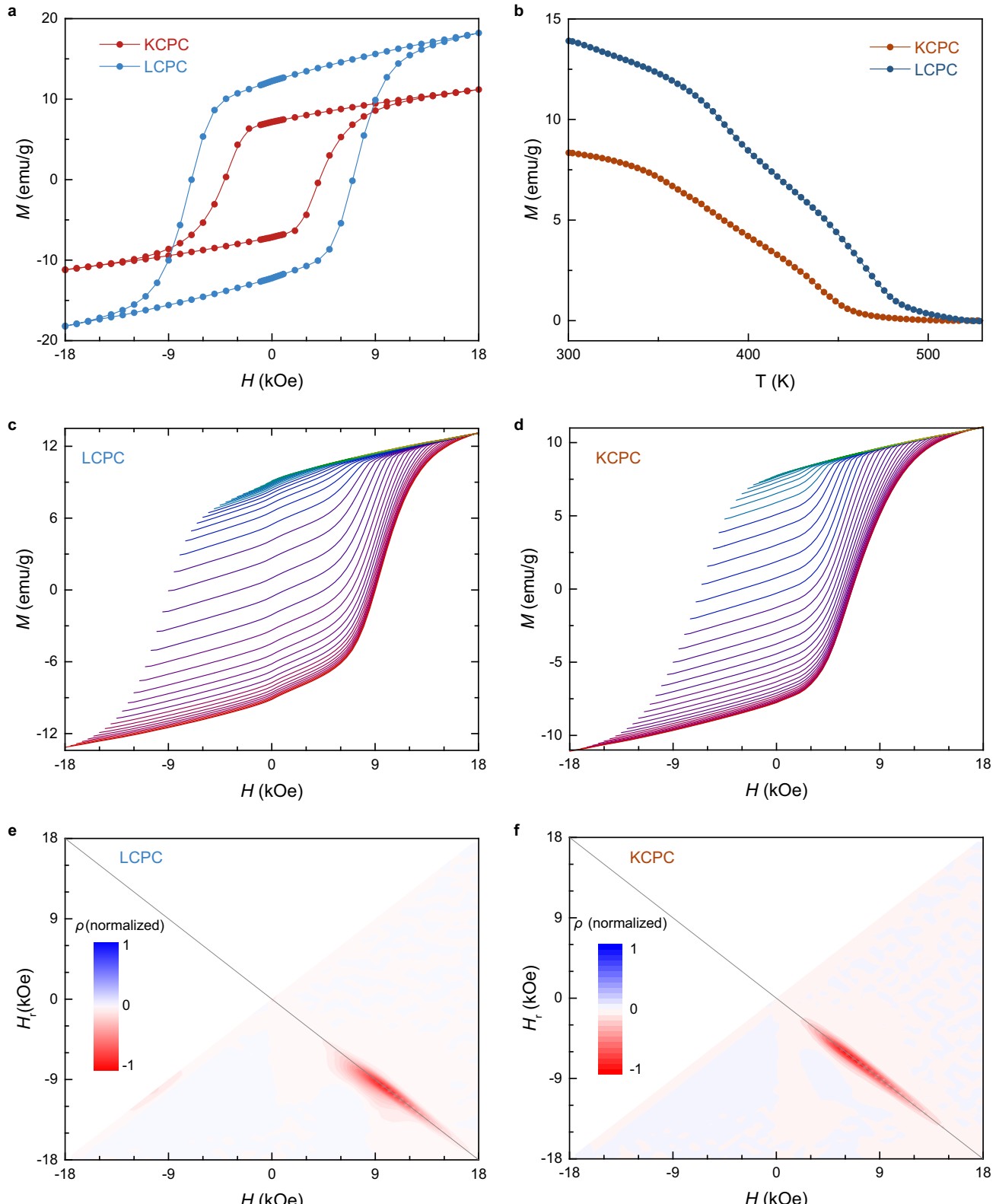

**Fig. 4 | Magnetic properties of LCPC and KCPC magnets. a** *M-H* loops of LCPC and KCPC magnets at room temperature. **b** Temperature-dependent magnetization measured at 5 kOe indicates magnetic ordering temperature in LCPC is higher than that in KCPC. **c**, **d** First-order reversed curves (FORC) of LCPC and KCPC magnets at room temperature. **e**, **f** The FORC diagrams of LCPC and KCPC magnets. The grey line is corresponding to $H_r = -H$.

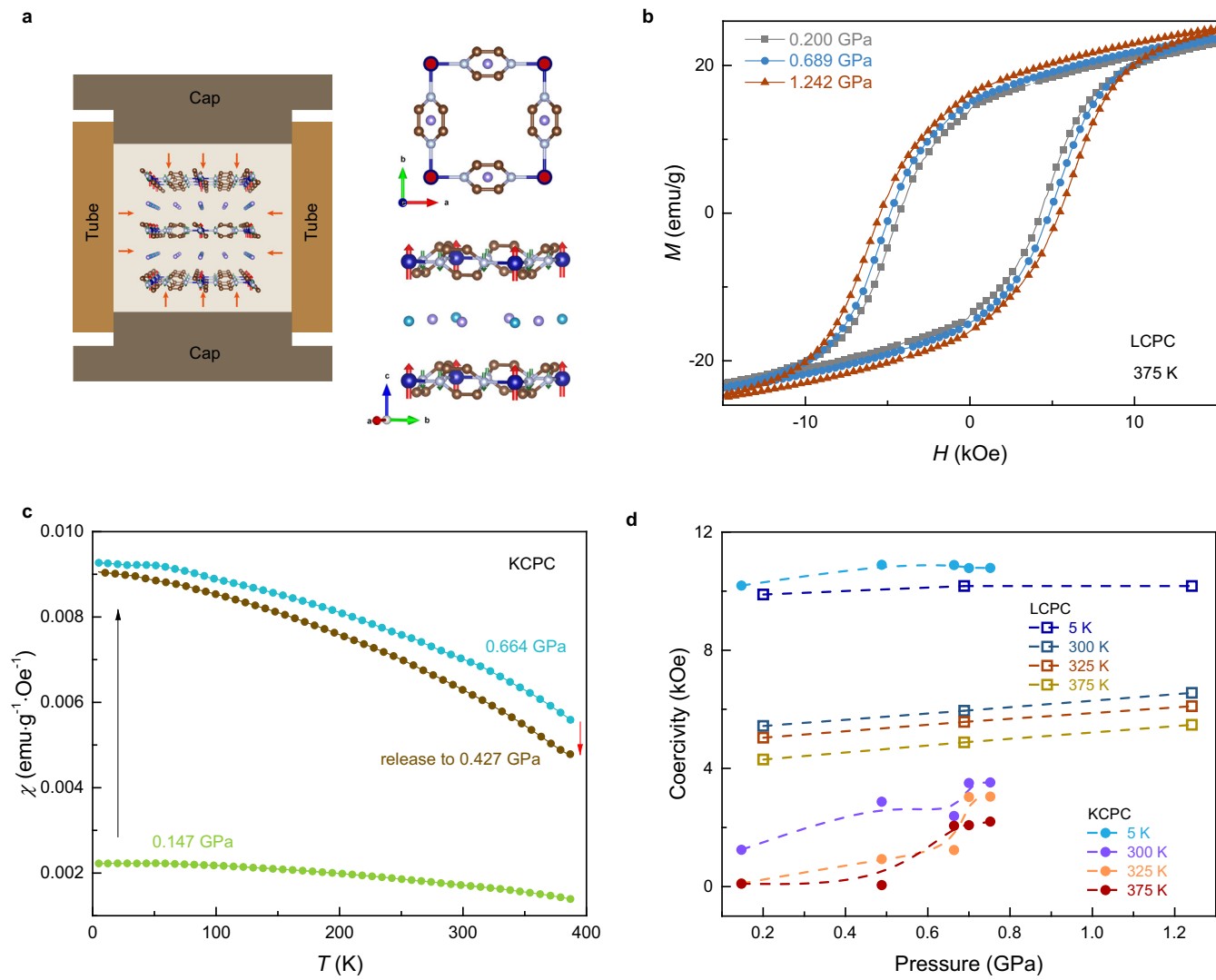

**Fig. 5 | Pressure effect on *M-H* loops of LCPC and KCPC magnets. a** The pressure cell setup is used for applying hydrostatic pressure on the layered magnets. The crystal structure shows the views along *c*-axis and in-plane directions. **b** The pressure effect on LCPC shows enhanced magnetism by increasing hydrostatic pressure. **c** Temperature-dependent magnetic susceptibility of KCPC shows enhanced magnetism. **d** The pressure-dependent coercivity of LCPC and KCPC magnets.

decreased *d*-spacing from the pressure effect. The peaks shift and the similar profile pattern from WAXS suggests the same crystal symmetry (tetragonal, P4/mmm) in KCPC samples before and after pressure.

Raman spectroscopy measurement was conducted on pristine KCPC bulk sample and the one after hydrostatic pressure, as shown in Fig. 6d. The stretching ($v_{ring}$) and in-plane bending ($\delta_{ring}$) modes of pyrazine ring shift to lower frequencies, while the in-plane bending of carbon-hydrogen bond ($\delta_{CH}$) and the stretching of pyrazine ring ($v'_{ring}$) shift to higher frequencies after pressure. The $\delta_{ring}$ and $v'_{ring}$ peaks are red- and blue-shifted by ~1.1 and 3.0 cm$^{-1}$, respectively. The hardened vibrational modes could be attributed by the shorter C-H bond length and smaller pyrazine ring in the compressed KCPC lattice, which is also consistent to the shift of WAXS peaks. In organic-based magnets, molecular arrangements can be largely tuned by pressure. Copper pyrazine dinitrate shows a pressure-induced structural transition[42], while pyrazine molecular crystal can change its molecular orientation in the crystal under pressure[43]. Therefore, the soft ligands of pyrazine molecules as magnetic superexchange pathways result in enhanced tunability in magnetism. In this context, first-principles simulations were conducted to reveal interlayer and intralayer pressure effect by compressing crystal lattice along *c* axis (mode A, uniaxial) or both *a*

and *b* axes (mode B, biaxial), respectively (Fig. 6e). In mode A, an enhanced magnetic moment of Cr-pyz layer is observed up to a compression ratio of 10% due to the decrease in magnetic moment of pyrazine, while the magnetic moment on Cr is almost unchanged (Fig. 6f and S16, 17). On the contrary, a decrease in magnetic moment of Cr-pyz layer occurs in mode B, indicating the interlayer pressure effect (mode A) is the dominant factor for the observed enhanced magnetic behavior in KCPC under pressure. The bond length of Cr-N slightly decreases in both modes A and B, implying the intramolecular deformation of pyrazine (Fig. 6g). Even if shortened by the compression, the distance of Cr-Cl still maintains long enough to avoid chemical reaction (Fig. S17). Therefore, the deformation of the soft ligand pyrazine plays a crucial role in determining the magnetism in KCPC under hydrostatic pressure.

## Discussion
The simulation results rule out the possibility of a pressure-induced structural transition in KCPC magnet. In our experiments, the hydrostatic pressure is below 1 GPa, corresponding to a compression ratio of ~2.2%[44]. In metal-pyrazine coordinated molecular magnets[42,45-47], the pressure-induced structural transition is usually at much higher

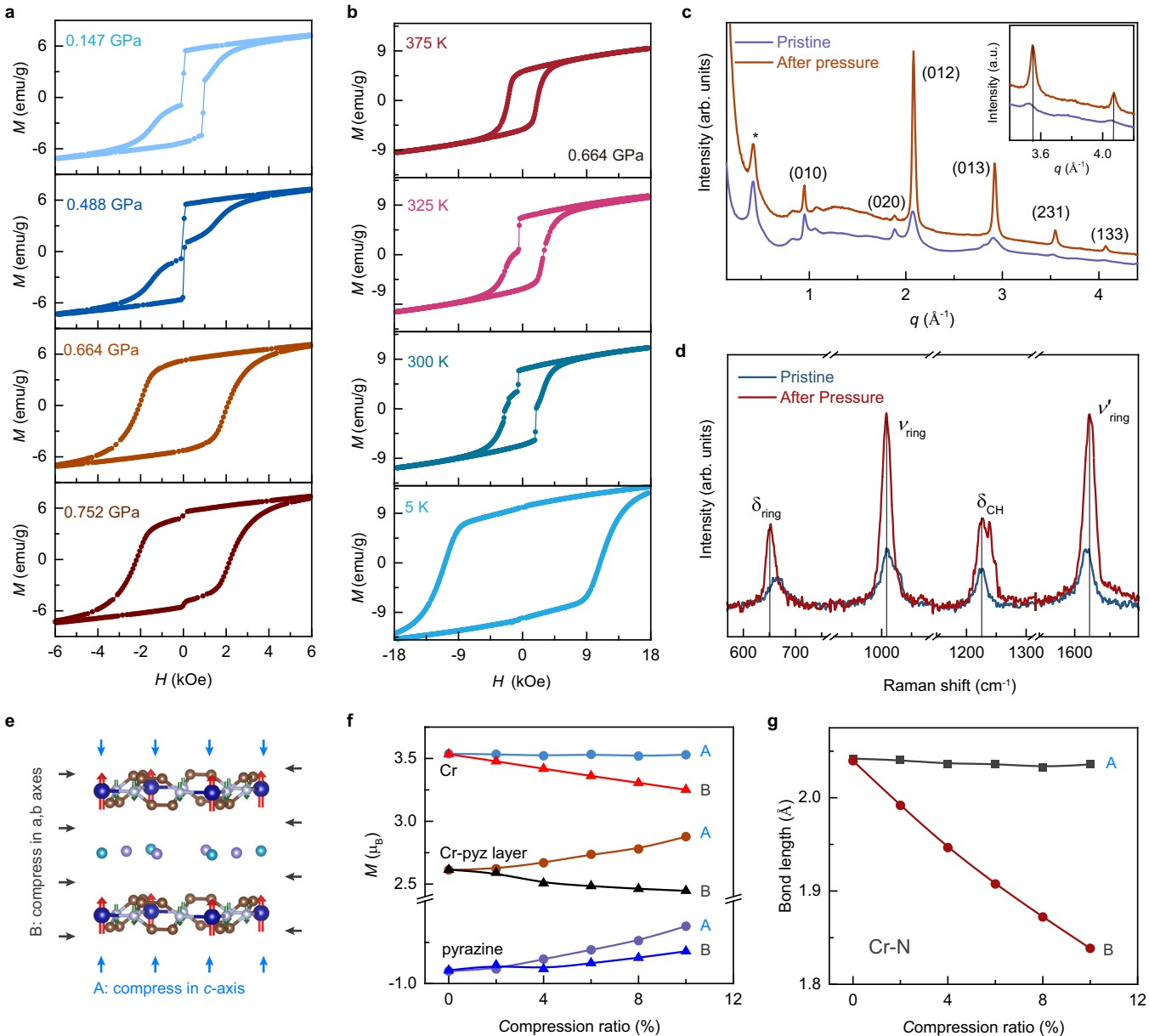

**Fig. 6 | Pressure-driven jumps in magnetization and the thermodynamic feature in KCPC magnet. a** The sharp jump at 375 K in *M-H* develops with increasing pressure from 0.147 GPa to 0.752 GPa. **b** *M-H* loops of KCPC magnet measured at 0.664 GPa indicate the temperature-dependent behavior of sharp jumps from 5 K to 375 K. **c** Wide-angle X-ray scattering patterns of KCPC magnet show the sharper and stronger peaks after pressure. The inset indicates the peak is shifting to a higher vector due to pressure effect. **d** Raman spectra of KCPC magnet before and after pressure. **e** The schematic presents the first-principles simulations by compressing in only *c*-axis (mode A) or *a,b* axes (mode B). **f** The compression ratio dependent magnetic moments of Cr, pyrazine, and Cr-pyz layer in two simulation modes A and B. **g** The bond length of Cr-N develops with compression ratio.

pressure, i.e., 6 GPa. The structural changes below 1 GPa often come from the modifications of bond length and angle without the symmetry change[42,45]. Therefore, according to our experimental and simulation results, the lattice change is suggested to induce the pressure effect on KCPC. The understanding of such sharp jumps in magnetization is unsettled, even though several mechanisms have been proposed for this behavior in different inorganic systems[48]. In the metamagnetic materials, magnetic field-dependent orbital ordering[49], martensitic-like transformation[50], and magnetic field-induced spin flop[51] are considered as possible origination of sharp jumps in magnetization. Here, the enhanced crystallinity of KCPC magnet by pressure may facilitate the possibility of magnetic field-induced spin flop. In conventional 2D inorganic magnets (like CrI₃)[2,12,52], layer-dependent studies on physical properties revealed ferromagnetism in monolayer

structure, and antiferromagnetic intralayer coupling in bilayer CrI₃[53]. Even in trilayer CrI₃, ferromagnetic order is restored. However, the key chemical composition, intra- and interlayer bonding, and structural differences between inorganic and metal-organic magnets present a significant challenge for layer-dependent studies in molecular-based magnets. It would be worth exploring a possible metamagnetic effect similar to that in bilayer CrI₃ from antiferromagnetic to ferromagnetic intralayer interaction[54]. Since atomically thin samples are very attractive for understanding their intrinsic magnetic order, such layer-dependent studies are still worth exploring once their large-size crystals are available for exfoliation.

In summary, room-temperature molecular layered magnets exhibit pressure-controlled magnetism with the coercivity coefficient $dH_c/dP$ up to 4 kOe/GPa. The stoichiometry and composition-

dependent magnetism reveal the alkali-metal reduction roles in the 2D metal-organic coordinated magnets. The coercivity and magnetization show the large pressure tunability in molecular layered magnet due to the increased magnetic coupling from the Cr-pyrazine interlayers and the deformation of soft ligand pyrazine molecule. In the layered KCPC magnet, spin crossover may occur as high-spin $Cr^{II}$ ($S = 2$) transition to low-spin $Cr^{II}$ ($S = 1$) state. Electron transfer could also be possible within the intra- and interlayer of Cr-pyrazine, while pressure induces charge redistribution between Cr and pyrazine by transferring charge from $Cr^{II}$ to pyrazine for the formation of $Cr^{III}$ ($S = 3/2$). This electron transfer can be facilitated through the structural transformation of $Cr^{III}$-Cl due to the pressure effect. The alkali-metal reduction and hydrostatic pressure pave a pathway toward the understanding of room-temperature 2D magnetism in molecule-based systems.

## Methods

### Preparation of LCPC and KCPC magnets
The solvothermal method[22,23] was used to prepare the LCPC and KCPC magnets. 231.3 mg 1,2-dihydroacenaphtylene and some lithium (for LCPC) or potassium (for KCPC) metals were mixed together by adding 8 ml tetrahydrofuran (THF) solvent. The mixture solution was stirred for three hours and then filtered. A 20 ml glass bottle was used to contain 5 ml THF solvent and 200 mg $Cr(pyz)_2Cl_2$ powder, as well as the filtered solution of alkali-metal cations. Then, the solution in the glass bottle was stirred for five days. The final product was cleansed and centrifuged twice with THF solvent, then kept in a vacuum chamber overnight to get dried LCPC and KCPC powders.

### Fourier-transform infrared spectra
Transmittance spectra of LCPC and KCPC magnets were collected on an Agilent Cary 630 FTIR spectrometer in an $N_2$ glovebox. The samples also were annealed at different temperatures and then used for FTIR measurements.

### Structural and morphologic characterizations
A Rigaku Ultima IV (40 kV, 44 mA) was used to characterize the crystal structures of LCPC and KCPC magnets by X-ray diffraction. The surface morphology was measured on Carl Zeiss AURIGA (200 kV) Field Emission Scanning Electron Microscope. The element analysis was determined by Oxford Energy-dispersive X-ray Spectrometer (EDS). JEOL JEM 2010 was used to collect the high-resolution transmission electron microscopic images and selected area electron diffraction patterns.

### Wide-angle X-ray scattering
X-ray scattering exposures in the wide-angle regime were performed on a Xenocs Xeuss 3.0 instrument. Exposures were obtained utilizing a Cu Kα microfocus source ($\lambda = 1.542$ Å) collimated with two sets of scatter less slits under vacuum. Sample to detector distance was calibrated using Silver (I) Behenate. SAXS patterns were recorded as a function of the scattering angle 2θ using a hybrid-pixel detector (Eiger 2R-1M) Samples were encapsulated in 1 mm disk washers and sealed with Kapton® (polyimide) film. Exposures were performed in transmission mode with additional exposures performed for an empty cell without a sample to allow for proper background subtraction of the scattering contribution by Kapton®. Exposures were obtained over a period of 30 seconds. 2D detector images were reduced to 1D scattering profiles via azimuthal averaging and corrected for detector geometry, flatfield, and beam transmission through the sample. Corrections for Bragg intensities (both polarization and absorption) were performed using data analysis tools in XSACT[55].

### Ultraviolet-visible-near-infrared spectroscopy measurements
Agilent Cary 7000 spectrophotometer was used to measure UV-Vis-NIR spectra of LCPC and KCPC magnets.

### In-situ M-H loop measurements
The Vibrating Sample Magnetometer (VSM, MicroSense EZ7-380V) VSM equipment with an open environment was used to conduct in-situ M-H loops measurements by using a sample holder with a reaction cell. The precursor $Cr(pyz)_2Cl_2$ and lithium solution are loaded into a reaction cell, which is immediately amounted onto a sample holder for measurement. A measurement sequence was run to repeatedly measure MH loops without changing measurement parameters. Each obtained M-H loop shows different features corresponding to its time order of completion. When the precursor $Cr(pyz)_2Cl_2$ and lithium solution were mixed in the reaction cell, the lithiation reaction occurs. The magnetic development of the precursor is monitored by continuous M-H measurement from the very beginning. Thus, we present the in-situ M-H loops of LCPC magnet, revealing the magnetic transformation during the reaction process.

### Ambient magnetic properties measurements
Ambient magnetic properties of LCPC and KCPC magnets were measured on a Vibrating Sample Magnetometer (VSM, MicroSense EZ7-380V) for high temperatures from 300 K to 530 K and a Physical Properties Measurement System EverCool II (PPMS, Quantum Design) for low temperatures from 8 K to 300 K. All the samples were sealed in epoxy and measured immediately. A room-temperature magnetism was still observed after 50 days (Fig. S18).

### Magnetic measurements under hydrostatic pressure
Pressure-dependent magnetization at different temperatures and magnetic field $M$ ($T, H$) were measured using a Physical Property Measurement System (PPMS Dynacool, Quantum Design, Inc. USA) with VSM option employing a Cu-Be cell manufactured by HMD (type CC-SPr-8.5D-MC4) with lead wire loaded together with the sample as an internal manometer. Daphne oil was used as a pressure-transmitting medium. The magnitude of applied hydrostatic pressure was determined by examining the superconducting transition temperature of lead. Magnetic hysteresis loops from 5 T to −5 T and temperature-dependent magnetic susceptibility from 5 K to 390 K were measured under different hydrostatic pressures.

### Spectroscopy measurements
Raman spectra of LCPC and KCPC magnets were collected on Renishaw inVia Raman Microscope. The samples were sealed in a quartz cube for protection. The excitation wavelength is 785 nm.

### Computational methods
All calculations were performed by using the Vienna Ab initio Simulation Package (VASP)[56] based on the density functional theory (DFT)[57,58]. We used the screened hybrid-functional of Heyd−Scuseria−Ernzerhof (HSE) with default mixing parameter of 25% and a standard range-separation parameter of 0.2 $Å^{-1}$[59,60]. To calculate the spin density near the nuclei, the projector-augmented-wave method (PAW)[61,62] and a plane-wave basis set were used. The plane-wave energy cutoff was set to 520 eV and a $3 \times 3 \times 3$ Γ centered k-point grid was used. Magnetic moments (in units of $\mu_B$) were computed by integrating the local spin densities on spheres around the atoms with Wigner-Seitz radii given by 1.588 Å, 1.323 Å, 0.370 Å, 0.863 Å, 0.741 Å, and 1.111 Å for K, Cr, H, C, N, and Cl, respectively, as implemented in VASP. The structural relaxations have been performed for all the systems investigated which were converged until the force acting on each ion was less than 0.1 eV/Å. The convergence criterion for total energy for structural relaxations was $10^{-5}$ eV.

## Data availability
All relevant experimental data are presented in the paper and the Supplementary Information. Additional data related to this paper can be provided by the corresponding author upon request.

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

## Acknowledgements

The U.S. Department of Energy, Office of Basic Energy Sciences, Division of Materials Sciences and Engineering supports S.R. under Award DE-SC0023433. The work at Buffalo State was supported by the National Science Foundation Award No. DMR-2213412. J.Y.T. and Q.M.Y. are supported by the National Science Foundation under Grant No. 2144936.

## Author contributions

Y.L.H. and S.Q.R. conceived the idea and designed the study. Y.L.H. conducted all sample syntheses and processing, magnetic measurements at ambient pressure, structural characterizations, X-ray diffraction, Raman and Fourier-transform spectroscopies. A.K.P. and N.K. conducted hydrostatic pressure measurements of magnetic properties. C.R. and M.T. contributed to wide-angle X-ray scattering. J.Y.T. and Q.M.Y. did the first-principles simulations. M.I. helped magnetic measurements. Y. H. helped with in-situ magnetic measurement. Y.L.H. and S.Q.R. wrote the manuscript with comments and inputs from all authors.

## Competing interests

The authors declare no competing interests.
