## [Peer Review File · Nature Communications]

Pressure-Controlled Magnetism in 2D Molecular LayersREVIEWERS' COMMENTS:

Reviewer #1 (Remarks to the Author):

I have carefully read the work by Y. Huang, et al. In this paper, they synthesized layered metal-organic magnets LCPC and KCPC and studied their response to applied pressure. In my opinion, the data is clearly presented and the interpretation is likely correct. The results may generate sufficient interest in the chemistry, materials science, and physics communities to warrant publication in Nature Communications. With that being said, I have a few questions and comments:

- 1) The systems studied here feature a layered 2D magnetic structure, which the authors claim is vital for the large reported pressure tunability. However, it is not immediately clear why these molecular systems should have much greater tunability (which is demonstrated) than conventional layered 2D magnetic systems, e.g. van der Waals crystals. The authors could expand upon the unique advantages offered by the molecular magnets in this regard, especially given the broad readership of Nat. Comm.
- 2) Additional investigation on the origin of the pressure-enhanced magnetism and sharp jumps in KCPC would be very useful. For example, probes like Raman could check whether the crystal structure changes with pressure. Especially since the authors claim "pressure-controlled peculiar magnetism through charge redistribution and structural transformation" without providing much evidence that the applied pressure indeed drives such processes, or what the dominant effect is here. If more experimental evidence is not possible, a more concrete discussion or insight is called for.
- 3) Were the sharp magnetic domains repeated in multiple KCPC samples? How do they vary from sample to sample?
- 4) In Fig. 6, M-H curves at select temperatures are shown, but there is no data shown between 300 – 5 K. Do the authors have a more complete temperature dependence data set they could include to show how the domain effect evolves from the sharp jump to smooth curve?

Other Comments:

- 1) The TEM images in Fig. 1b and Supplementary Fig. S4c appear to be the exact same image.
- 2) The authors present the pressure tunability coefficient in Fig. 1c before any pressure dependent data is shown. While certainly one of the main findings of the paper, in this humble reviewer's opinion the pressure data should come before the summary/comparison plot of the results.
- 3) In Fig. 1 caption, it says "The reduced lattice (dash) indicates the pressure effect on crystal structure, resulting in a closer interlayer spacing." I do not see any dash in the included manuscript.
- 4) There are quite a few typos: for example "Barhausen" on page 10.

Reviewer #2 (Remarks to the Author):

The authors report the pressure control of interlayer magnetic coupling in molecular layer compounds. Two kinds of layered metal-organic magnets are prepared and characterized using several experimental techniques. The key finding here is the pressure tuning of magnetic coercivity. This pressure effect is discussed by defining the coercivity coefficient as the pressure derivative of magnetic coercivity, and it is calculated to be 4 kOe/GPa which is quite large. However, these observations are closely related to the pressure-induced sharp jumps in the magnetization (Fig. 6), and the explanations for them are not clear. Although the authors mentioned this "can be regarded as the Barkhausen jump", there are not enough experiments and discussions about its physical origins and the underlying mechanism. Therefore, I cannot recommend it for publication, at least in the present form.

Reviewer #3 (Remarks to the Author):

This paper reports the pressure control of magnetism in two-dimensional layered metal-organic compounds with a high T_c of around 500 K. The authors used PPMS to examine the magnetism of bulk crystals under hydrostatic pressure. A definitive M-H loop was detected. The authors also state that the magnetic properties can be enhanced by hydrostatic pressure. However, establishing a rigorous criterion to judge the magnetism of 2D layered materials is necessary. Unfortunately, the links between nano-micro/ molecular structure and magnetism are too weak. To some extent, the present results can not prove that the enhanced magnetism originate from the interlayer coupling. The pressure can also enhance the intralayer coupling of Cr atoms. Furthermore, the authors do not study and demonstrate the interlayer coupling of the 2D layer materials. Therefore, publication is not recommended. Here are some of my concerns and comments to the authors.

(1) What's the interlayer coupling of magnetism? AFM or FM style? Is it similar to typical 2D magnetism CrI₃? The authors must exfoliate mono-, bi- and tri-layer materials to investigate the layer-dependence of magnetism by scanning RMCD or MOKE, moreover, the magnetic domains should be observed.

(2) The Raman and FTIR spectra clearly show the features of 2D layer metal-organic compounds. Does the pressure influence these features? The in-situ RMCD and Raman measurements under pressure should be used to validate the relations of molecular structures and magnetism, in particular, monolayer and bilayer samples can confirm whether the enhancement arise from enhanced intralayer or interlayer coupling.

(3) The authors assigned the sharp jumps in magnetization to the Barkhausen jump due to the rapid changes of the size and orientation of magnetic domain. The explanation is plausible. The sharp jumps appear at 0.147 GPa and disappear at 0.664 GPa, and similar behavior also can be observed in temperature dependence of M-H loops. How does the pressure tune the interlayer/intralayer coupling and magnetic domains? The present experimental results are not enough. Layer dependence of RMCD mapping must be done.

(4) How do the authors do the in-situ M-H loops of LCPC magnet during the solution reaction (Fig. 2b)? Was the solution reaction cell loaded into PPMS?

RESPONSE TO REVIEWERS' COMMENTS

Reviewer #1 (Remarks to the Author):

I have carefully read the work by Y. Huang, et al. In this paper, they synthesized layered metal-organic magnets LCPC and KCPC and studied their response to applied pressure. In my opinion, the data is clearly presented and the interpretation is likely correct. The results may generate sufficient interest in the chemistry, materials science, and physics communities to warrant publication in Nature Communications. With that being said, I have a few questions and comments:

1) The systems studied here feature a layered 2D magnetic structure, which the authors claim is vital for the large reported pressure tunability. However, it is not immediately clear why these molecular systems should have much greater tunability (which is demonstrated) than conventional layered 2D magnetic systems, e.g. van der Waals crystals. The authors could expand upon the unique advantages offered by the molecular magnets in this regard, especially given the broad readership of Nat. Comm.

Reply: We thank the reviewer for the insightful comment. Molecular magnets or organic-based magnets show unique advantages of lightweight, low-temperature processability, and synthetic controlled structural/magnetic property tailoring, in which the spin coupling often occur among transition metals and molecular ligands as metal-organic frameworks. The d electron from transition metal and π electron from organic molecule lead to the d- π interaction in the metal-organic framework. Dynamic electronic structure phenomena including spin crossover, charge transfer and linkage isomerization are also known in molecular magnets, showing the large tunability under external stimuli of pressure, illumination, and thermal activation. A long-standing missing member in the family of 2D magnets, e.g. van der Waals (vdW) crystals, is 2D molecular hard magnets that exhibit large magnetic anisotropy and operate above room temperature. Conventional layered 2D magnets have shown pressure dependent tunability in lattice structure (stacking arrangement or interlayer spacing), resulting in metamagnetic transitions. For example, in bilayer CrI₃, a pressure-induced phase transition is shown from antiferromagnetic to ferromagnetic order¹. Even in a trilayer CrI₃, one ferromagnetic phase and two types of antiferromagnetic phases could be induced due to the changes in stacking arrangement under pressure. In 2D molecular magnets, metal-organic networks could also exhibit the tunability in lattice structure as shown in the conventional inorganic 2D magnets. Besides, molecules in 2D organic-based magnets can present different oriented angles in the layer, a variable charge state, as well as a drastic compression ratio. Therefore, molecular structural building blocks in 2D molecular based magnets could possess the unique stimuli-responsive features.

2) Additional investigation on the origin of the pressure-enhanced magnetism and sharp jumps in KCPC would be very useful. For example, probes like Raman could check whether the crystal structure changes with pressure. Especially since the authors claim “pressure-controlled peculiar magnetism through charge redistribution and structural transformation” without providing much evidence that the applied pressure

indeed drives such processes, or what the dominant effect is here. If more experimental evidence is not possible, a more concrete discussion or insight is called for.

Reply: We thank the reviewer for the helpful suggestion. Raman spectroscopy measurement was conducted on KCPC sample after hydrostatic pressure, as shown in Fig. R1. The stretching (ν_{ring}) and in-plane bending (δ_{ring}) modes of pyrazine ring shift to lower frequencies (Fig. R1a), while the in-plane bending of carbon-hydrogen bond (δ_{CH}) and the stretching of pyrazine ring (ν'_{ring}) shift to higher frequencies after pressure (Fig. R1b). The δ_{ring} and ν'_{ring} peaks are red- and blue-shifted by about 1.1 and 3.0 cm^{-1} , respectively. Raman peak shifts due to the pressure confirm the structural change in KCPC lattice. We also add this figure in the Fig. 6c-d in the revised manuscript.

Figure R1. Raman spectra of KCPC magnet before (black) and after pressure (red).

3) Were the sharp magnetic domains repeated in multiple KCPC samples? How do they vary from sample to sample?

Reply: We thank the reviewer for this question. The sharp jump is observed in multiple KCPC samples. As plotted in below Fig. R2, the sample under 0.163 GPa shows sharp jumps at $H_1 = -854$ Oe and $H_2 = 305$ Oe when temperature is 350 K. This MH-loop is a little different since magnetization is reversible after such sharp jumps. Most observed results are similar to the M-H loop at 325 K. Such difference may be attributed to the sample stoichiometry because it's not exactly getting the same stoichiometric sample from a two-step reaction with solid precursor and lithium solution process. We mention the repeatable sharp jumps in magnetization in the revised manuscript, but also notice that the stoichiometry study would be valuable.

Figure R2. M-H loops of KCPC sample under 0.163 GPa.

4) In Fig. 6, M-H curves at select temperatures are shown, but there is no data shown between 300 – 5 K. Do the authors have a more complete temperature dependence data set they could include to show how the domain effect evolves from the sharp jump to smooth curve?

Reply: We thank for this comment. We focused on the M-H loops at above 300 K since this material magnetically order at high temperature (T_c : 510K). As a follow-up of this comment, the low-temperature M-H loops were measured and supplemented in the supporting information.

Figure R3. M-H loops of KCPC sample measured at (a) 0.330 GPa and (b) 0.752 GPa.

Other Comments:

1) *The TEM images in Fig. 1b and Supplementary Fig. S4c appear to be the exact same image.*

Reply: Thank reviewer for this reminder. The TEM image in Fig. S4c has been replaced with a new image in the revised manuscript.

2) *The authors present the pressure tunability coefficient in Fig. 1c before any pressure dependent data is shown. While certainly one of the main findings of the paper, in this humble reviewer's opinion the pressure data should come before the summary/comparison plot of the results.*

Reply: We appreciate the reviewer's suggestion and we surely understand the reviewer's kind opinion. The presentation of pressure coefficient in Fig. 1c is aimed to introduce the main finding of pressure effect on coercivity. Given the crystal structure with spin assignment on Cr atoms and pyrazine molecules in Fig. 1a, the pressure coefficients of coercivity clearly show the large magnetic response in both LCPC and KCPC magnets. With these results as well as the 2D structure, it's more efficient to highlight pressure effect on 2D metal-organic magnet.

3) *In Fig. 1 caption, it says "The reduced lattice (dash) indicates the pressure effect on crystal structure, resulting in a closer interlayer spacing." I do not see any dash in the included manuscript.*

Reply: We apologize for this misleading sentence that has been deleted in the revised manuscript.

4) *There are quite a few typos: for example "Barhausen" on page 10.*

Reply: We thank the reviewer for the careful review. We have corrected the misspelling regarding the term "Barkhausen" as well as other typos.

Reviewer #2 (Remarks to the Author):

The authors report the pressure control of interlayer magnetic coupling in molecular layer compounds. Two kinds of layered metal-organic magnets are prepared and characterized using several experimental techniques. The key finding here is the pressure tuning of magnetic coercivity. This pressure effect is discussed by defining the coercivity coefficient as the pressure derivative of magnetic coercivity, and it is calculated to be 4 kOe/GPa which is quite large. However, these observations are closely related to the pressure-induced sharp jumps in the magnetization (Fig. 6), and the explanations for them are not clear. Although the authors mentioned this “can be regarded as the Barkhausen jump”, there are not enough experiments and discussions about its physical origins and the underlying mechanism. Therefore, I cannot recommend it for publication, at least in the present form.

Reply: We really thank the reviewer for the careful assessment and helpful comment on our manuscript. A pressure-induced sharp jump in the magnetization could originate from several mechanisms according to different systems. The sharp jumps in the magnetization correspond to the behavior of metamagnetic phase transition, which is often induced by magnetic field, temperature, pressure and other external perturbations. For example, a magnetic-field-induced spin flop transition was reported in multiferroic $\text{Ca}_3\text{CoMnO}_6$ ². Besides, field-induced orbital ordering, martensitic-like transformation, spin quantum transition, geometric frustration and spin reorientation have been proposed to understand the metamagnetic phase transition in inorganic systems. In organic-based magnets, molecular arrangements can be largely tuned by pressure. Copper pyrazine dinitrate shows a pressure-induced structural transition³. Pure pyrazine molecular crystal can change its molecular orientation in the crystal under pressure⁴. Therefore, the soft ligands of pyrazine molecules as magnetic superexchange pathways result in enhanced tunability in magnetism. We also addressed this discussion in the revised manuscript.

Reviewer #3 (Remarks to the Author):

This paper reports the pressure control of magnetism in two-dimensional layered metal-organic compounds with a high T_c of around 500 K. The authors used PPMS to examine the magnetism of bulk crystals under hydrostatic pressure. A definitive M-H loop was detected. The authors also state that the magnetic properties can be enhanced by hydrostatic pressure. However, establishing a rigorous criterion to judge the magnetism of 2D layered materials is necessary. Unfortunately, the links between nano-micro/ molecular structure and magnetism are too weak. To some extent, the present results can not prove that the enhanced magnetism originate from the interlayer coupling. The pressure can also enhance the intralayer coupling of Cr atoms. Furthermore, the authors do not study and demonstrate the interlayer coupling of the 2D layer materials. Therefore, publication is not recommended. Here are some of my concerns and comments to the authors.

(1) What's the interlayer coupling of magnetism? AFM or FM style? Is it similar to typical 2D magnetism CrI_3 ? The authors must exfoliate mono-, bi- and tri-layer materials to investigate the layer-dependence of magnetism by scanning RMCD or MOKE, moreover, the magnetic domains should be observed.

Reply: Thanks for the reviewer's comment on our manuscript. In the Cr-pyrazine layers, the local magnetic moments of the pyrazine ligands ($S = \frac{1}{2}$, pyz^-) are antiparallel to the nearest Cr moments ($S = 2$, Cr^{2+}), constructing a ferrimagnetic order. The spacing layers partially occupied by LiCl or KCl show non-magnetic order. The key chemical composition, intra- and inter-layer bonding, and structural differences between inorganic and metal-organic magnets present a significant contrast for the layer-dependent studies in molecular based magnets. In this study, we focused on macroscopic magnetism in LCPC and KCPC under hydrostatic pressure by using all accessible techniques to reveal their room-temperature magnetic order. As a follow up to the insightful comments from the reviewer, we have done additional studies on several-layer LCPC and KCPC magnets which are attached below, and in addition, we provide the justification below. We also add the discussion and literature references accordingly into the revised manuscript.

In the inorganic magnet chromium triiodide CrI_3 , a ferromagnetic order is established below 61 K with out-of-plane spin orientation. The monolayer CrI_3 still shows ferromagnetic order with Curie temperature of 45 K, while the bilayer CrI_3 shows suppressed magnetization with a metamagnetic effect by magnetic field¹. Besides, the ferromagnetic order is restored in trilayer CrI_3 . The study of layer-dependent magnetic phase in CrI_3 is based on the large-sized crystal samples⁵. As shown in Fig. R4, the CrI_3 single crystals can be grown in a large size (> 1 cm) and with a flat surface, facilitating the accessibility of exfoliation for mono-, bi-, and trilayer structures. The CrI_3 single crystals can maintain high quality after a long-time exposure of air, ensuring the stability during exfoliation. Those advantages for exfoliation cannot be accessed in the metal-organic magnets.

Instead of a large-sized ionic-bonded inorganic crystal, the metal-organic framework magnets, KCPC and LCPC in this study, are constructed from metal-center

and organic ligand interactions which are grown in an average particle size of 1~2 μm (Fig. R4d). Such molecule-based magnets are grown by post-synthetic reduction of precursor powder sample at the solid-liquid interface, which differentiate from the one-step chemical vapor transport method for inorganic materials. Given a sample size four orders of magnitude less than that of CrI_3 , the powder samples of metal-organic magnets LCPC and KCPC present a significant challenge to be exfoliated for the layer-dependent studies. In addition, with the chemical element of alkali metal, LCPC and KCPC are sensitive to environment, limiting the access of its processability. Furthermore, the intralayer ionic interaction in LCPC and KCPC is much stronger than the vdW interaction in 2D inorganic magnets. All these contribute to the challenges encountered in the studies of metal-organic magnets in the literature studies, while primarily focusing on the macroscopic magnetism and its stimuli dependent studies. As a follow-up, we are currently investigating the one-step synthetic routes by using large-sized precursor $\text{Cr}(\text{pyz})_2\text{Cl}_2$ with the objective to achieve large crystals of LCPC and KCPC for future potential layer-dependent studies.

Figure R4. Crystal structures and sample morphologies of CrI_3 and $\text{A}_x\text{Cr}(\text{pyz})_2\text{Cl}_2$ molecular magnets. (a)-(c) X-ray diffraction pattern, optical image and crystal structure of CrI_3 . These results come from the work of Michael A. McGuire *et al.*⁵ (d)-(e) XRD pattern, SEM image and crystal structure of $\text{A}_x\text{Cr}(\text{pyz})_2\text{Cl}_2$ ($\text{A} = \text{Li}$ and K).

(2) The Raman and FTIR spectra clearly show the features of 2D layer metal-organic compounds. Does the pressure influence these features? The in-situ RMCD and Raman measurements under pressure should be used to validate the relations of molecular structures and magnetism, in particular, monolayer and bilayer samples can confirm whether the enhancement arise from enhanced intralayer or interlayer coupling.

Reply: We thank the reviewer for the helpful comment. The structure of metal-organic magnets indeed relates to magnetism under hydrostatic pressure. Even though we have not accessed to monolayer and bilayer samples due to the challenges mentioned above, we conducted Raman spectroscopy measurement is conducted on KCPC bulk sample before and after hydrostatic pressure, as shown in Fig. R5. The stretching (ν_{ring}) and in-plane bending (δ_{ring}) modes of pyrazine ring shift to lower frequencies (Fig. R5a), while the in-plane bending of carbon-hydrogen bond (δ_{CH}) and the stretching of pyrazine ring (ν'_{ring}) shift to higher frequencies after pressure (Fig. R5b). The δ_{ring} and ν'_{ring} peaks are red- and blue-shifted by about 1.1 and 3.0 cm^{-1} , respectively. Raman peak shifts due to the pressure confirm the structural change in KCPC lattice. We also added this figure in the Fig. 6c-d in the revised manuscript.

Figure R5. Raman spectra of KCPC magnet before (black) and after pressure (red).

(3) The authors assigned the sharp jumps in magnetization to the Barkhausen jump due to the rapid changes of the size and orientation of magnetic domain. The explanation is plausible. The sharp jumps appear at 0.147 GPa and disappear at 0.664 GPa, and similar behavior also can be observed in temperature dependence of M-H loops. How does the pressure tune the interlayer/intralayer coupling and magnetic domains? The present experimental results are not enough. Layer dependence of RMCD mapping must be done.

Reply: We thank the reviewer for reminding of the very interesting topic of pressure effect on interlayer/intralayer coupling and magnetic domains. The current challenges of exfoliation on powder samples hinder the further exploration on achieving atomically thin molecular metal-organic magnets. We have added experiments to mechanically exfoliate the powder sample as shown in Fig. R6, though promising results haven't been achieved so far due to the challenges of molecular based magnets posted.

Figure R6. (a) Optical image of a quartz slide after sample transfer. The inset is the plot of Raman spectra. (b) Optical image of a scotch tape after mechanically exfoliating KCPC powder. The inset shows the powder cluster. Layered samples are hardly seen at such scale. (c)-(f) Optical images of KCPC sample clusters on silicon substrates after exfoliation. (g) Raman spectra of those sample clusters in (c)-(f), compared with the result of bulk KCPC.

(4) How do the authors do the in-situ M-H loops of LCPC magnet during the solution reaction (Fig. 2b)? Was the solution reaction cell loaded into PPMS?

Reply: Thanks for your reminder. We have added the detailed experimental methods and protocols into the revised main text. The VSM equipment with an open environment is used to conduct in-situ M-H loops measurements by using a sample holder with a reaction cell. The precursor $\text{Cr}(\text{pyz})_2\text{Cl}_2$ and lithium solution are loaded into a reaction cell, which is immediately amounted onto a sample holder for measurement. A measurement sequence was run to repeatedly measure MH loops without changing measurement parameters. Each obtained M-H loop shows different features corresponding to its time order of completion. When the precursor $\text{Cr}(\text{pyz})_2\text{Cl}_2$ and lithium solution were mixed in the reaction cell, the lithiation reaction occurs. The magnetic development of the precursor is monitored by continuous M-H measurement from the very beginning. Thus, we present the in-situ M-H loops of LCPC magnet, revealing the magnetic transformation during reaction process.

References

1. Huang, B. *et al.* Layer-dependent ferromagnetism in a van der Waals crystal down to the monolayer limit. *Nature* **546**, 270-273 (2017).
2. Flint, R., Yi, H.T., Chandra, P., Cheong, S.W. & Kiryukhin, V. Spin-state crossover in multiferroic $\text{Ca}_3\text{Co}_{2-x}\text{Mn}_x\text{O}_6$. *Phys. Rev. B* **81**, 092402 (2010).
3. O'Neal, K.R. *et al.* Pressure-induced structural transition in copper pyrazine dinitrate and implications for quantum magnetism. *Phys. Rev. B* **93**, 104409 (2016).
4. Maehara, M., Kawano, H., Nibu, Y., Shimada, H. & Shimada, R. Pressure Effect on the Inter- and Intramolecular Vibrations of Pyrazine Crystal. *Bulletin of the Chemical Society of Japan* **68**, 506-511 (1995).
5. McGuire, M.A., Dixit, H., Cooper, V.R. & Sales, B.C. Coupling of Crystal Structure and Magnetism in the Layered, Ferromagnetic Insulator CrI_3 . *Chem. Mater.* **27**, 612-620 (2015).

REVIEWER COMMENTS

Reviewer #1 (Remarks to the Author):

I thank the authors for the additional data and efforts they have provided. However, more rigorous evidence or reasoning is required before publication.

1) Specifically, the Raman spectra indeed shows pressure effects, but the interpretation of the data is unclear. For example, is the claimed "structural change" a change in the crystal symmetry (similar to e.g., CrI₃ which undergoes a stacking order phase transition), or simply a change in the lattice constants?

2) The challenge of disentangling the effects of changing intra-layer and inter-layer effects (raised by other Reviewers as well) has not been satisfactorily answered by the Raman data and analysis. For instance, the in-plane modes are also changed by the pressure. Therefore the origin of the observed effects is still unresolved. Considering this is a central claim of the paper, it must be well substantiated.

Reviewer #2 (Remarks to the Author):

I have read through the point-by-point response and the revised manuscript. Although some of the questions raised by other reviewers have been answered, I still believe the key claims and findings are not fully supported and explained. As the authors want to highlight in Fig. 1, the most important finding is the pressure tunability coefficient, which has a high value. Based on the authors' definition, this calculated high value mainly originates from the pressure-induced sharp jumps in the magnetization (Fig. 6). However, the observations are still vaguely discussed as a metamagnetic effect, while no additional experimental evidence is provided to discuss the underlying mechanism. Also as pointed out by the other reviewers, in general, the link between the structure and magnetism under pressure is still weak, and a more rigorous understanding of the pressure effects on both structure and magnetism is still missing. With these points being said, unfortunately, I still hold the opinion as my first report.

Reviewer #3 (Remarks to the Author):

The authors have well addressed my questions! I recommend it public in the journal!

RESPONSE TO REVIEWERS' COMMENTS

Reviewer #1:

I thank the authors for the additional data and efforts they have provided. However, more rigorous evidence or reasoning is required before publication.

1) Specifically, the Raman spectra indeed shows pressure effects, but the interpretation of the data is unclear. For example, is the claimed “structural change” a change in the crystal symmetry (similar to e.g., CrI₃ which undergoes a stacking order phase transition), or simply a change in the lattice constants?

Reply: We thank the reviewer’s insightful comments and inputs on the structural change under pressure. A change of lattice parameter is interpreted as the plausible factor to underline the pressure effects based on the additional experiments and first-principles simulations we carried out in this revision. In the following, we illustrated such proposed mechanism below for your consideration, which has been added into the revision as well.

First, a wide-angle X-ray scattering (WAXS) is carried out on the KCPC samples before and after pressure. The obtained peaks (012), (013), (231), (133) in KCPC after pressure are shifting to a higher scattering vector q compared to those of the pristine sample (Fig. R1), revealing a decreased d -spacing from the pressure effect. The lattice of KCPC is compressed under hydrostatic pressure, while remaining after releasing pressure. In addition, the sharper and stronger peaks in the KCPC after pressure indicate a higher crystallinity due to the pressure effect. There is no new structural phase found in the KCPC after pressure. The peaks shift and the similar profile pattern suggest the same crystal symmetry (tetragonal, P4/mmm) in KCPC samples before and after pressure.

Second, Raman spectroscopy measurements indicate that the in-plane bending of carbon-hydrogen bond (δ_{CH}) and the stretching of pyrazine ring (ν'_{ring}) shift to higher frequencies after pressure (Fig. R2). The hardened vibrational modes could be attributed by the shorter C-H bond length and smaller pyrazine ring in the compressed KCPC lattice, which is also consistent to the shift of WAXS peaks.

In addition, first-principles simulations reveal an enhanced magnetic moment of Cr-pyz layer due to the decrease in magnetic moment of pyrazine when c -axis is compressed by a ratio up to 12% (Fig. R3a-b), while the Cr moment almost maintains unchanged. By pressure, the bond length of Cr-N slightly decreases, indicating the intramolecular deformation of pyrazine (Fig. R3c). The shortened distance of Cr-Cl still maintains long enough to avoid chemical reaction (Fig. R3d). The persisting stability of the initial crystal symmetry is confirmed after the optimization of compressed crystal structure.

The structural transition does not appear under the simulation even of 12% compression ratio. In our experiments, the hydrostatic pressure is below 1 GPa, corresponding to a compression ratio of $\sim 2.2\%$ ¹. In metal-pyrazine coordinated molecular magnets,^{1, 2, 3, 4, 5} the pressure induced structural transition is usually at much higher pressure, i.e. 6 GPa. The structural changes below 1 GPa often come from the modifications of bond length

and angle without the symmetry change.^{2, 3} Therefore, according to our experimental and simulation results, the lattice change is suggested to be induce the pressure effect on KCPC.

Figure R1. Wide-angle X-ray scattering (WAXS) patterns of KCPC samples before and after pressure. (a) The whole profiles of WAXS patterns show sharper and stronger peaks after pressure. (b-e) Peaks comparison indicates the shift to higher scattering vector.

Figure R2. Raman spectra of KCPC magnet before (black) and after pressure (red). (a) The stretching (ν_{ring}) and in-plane bending (δ_{ring}) modes of pyrazine ring shift to lower frequencies. (b) The in-plane bending of carbon-hydrogen bond (δ_{CH}) and the stretching of pyrazine ring (ν'_{ring}) shift to higher frequencies after pressure.

Figure R3. First-principles simulation of magnetism and structural parameters of KCPC with c -axis compression ratio up to 12%. (a) The magnetic moment of pyrazine molecule decreases in value with a compressed c -axis constant. (b) The magnetic moment of Cr is almost unchanged, while magnetic moment of Cr-pyz layer increases continuously. (c) Bond length of Cr-N in pyrazine molecule decrease, as well as the distance of Cr---Cl distance.

2) *The challenge of disentangling the effects of changing intra-layer and inter-layer effects (raised by other Reviewers as well) has not been satisfactorily answered by the Raman data and analysis. For instance, the in-plane modes are also changed by the pressure. Therefore the origin of the observed effects is still unresolved. Considering this is a central claim of the paper, it must be well substantiated.*

Reply: We appreciate the reviewer's nice inputs in the intra-layer and inter-layer effects. Under pressure, the parameters of crystal structure in intra-layer and inter-layer directions change, which leads to the varied interactions in both intra- and inter-layer. The first-principle simulations are carried out to resolve which interaction is dominant under pressure.

When c -axis parameter is compressed, the negative magnetic moment of pyrazine molecule decreases in value to induce an enhancement of magnetism in Cr-pyz layer (Fig. R3a-b). The distance between Cr and pyrazine is relatively enlarged (Fig. R3c-d). Therefore, pressure induces a magnetic decrease of pyrazine in the antiferromagnetically coupled Cr-pyrazine layer, because of the decreased interlayer spacing. The simulated pressure-induced metamagnetic transition results in an enhanced magnetic moment that is consistent with the experimental result under pressure as shown in Fig. 5c in the manuscript.

In order to clarify the intralayer effect, $a(b)$ -axis parameter is compressed, while c -axis constant is kept unchanged. A decrease of magnetism in Cr-pyz layer is induced by the magnetic decrease in both Cr and pyrazine (Fig. R4a-b). The distance between Cr and pyrazine is largely reduced when c -axis constant is not changed (Fig. R3c-d). Therefore,

intralayer effect induced by pressure decreases the magnetic moment of KCPC, which is not present in our experimental results. At least, the intralayer effect is much weak so that the interlayer effect for magnetic enhancement can be dominant.

Figure R4. First-principles simulation of magnetism and structural parameters of KCPC with a -axis compression ratio up to 16%. (a) The magnetic moment of pyrazine molecule decreases in value with a compressed a -axis constant. (b) The magnetic moment of Cr continuously decreases with compression, while magnetic moment of Cr-pyz layer decreases first and then shows a little upturn. (c) Bond length of Cr-N in pyrazine molecule decrease, while Cr---Cl distance is kept unchanged as c -axis is not compressed.

Reviewer #2:

I have read through the point-by-point response and the revised manuscript. Although some of the questions raised by other reviewers have been answered, I still believe the key claims and findings are not fully supported and explained. As the authors want to highlight in Fig. 1, the most important finding is the pressure tunability coefficient, which has a high value. Based on the authors' definition, this calculated high value mainly originates from the pressure-induced sharp jumps in the magnetization (Fig. 6). However, the observations are still vaguely discussed as a metamagnetic effect, while no additional experimental evidence is provided to discuss the underlying mechanism. Also as pointed out by the other reviewers, in general, the link between the structure and magnetism under pressure is still weak, and a more rigorous understanding of the pressure effects on both structure and magnetism is still missing. With these points being said, unfortunately, I still hold the opinion as my first report.

Reply: We thank the reviewer for the nice comments on the underlying mechanism of the sharp jumps in the magnetization and the link between the structure and magnetism. As a follow-up, we carried out additional experiments and simulation modeling (first-principle simulations) to provide an understanding for the evolution of magnetism under pressure as well as the relationship between structure and magnetism. As c-axis parameter is compressed, the magnetic moment of pyrazine molecule changes to induce an enhancement of magnetism in Cr-pyz layer (Fig. R3a-b), while the distance between Cr and pyrazine is relatively enlarged (Fig. R3c-d). Therefore, pressure could induce a magnetism decrease of pyrazine in the antiferromagnetically coupled Cr-pyrazine layer. The pressure-induced metamagnetic transition results in an enhanced magnetic moment that is consistent with the experimental result under pressure as shown in Fig. 5c in the revised manuscript. KCPC samples became compact and got a higher crystallinity under pressure.

We exclude a structural transition for the metamagnetic behavior from the wide-angle X-ray scattering (WAXS) and Raman spectroscopy measurements on KCPC samples before and after pressure. The WAXS peaks (020), (012), (013), (231), (133) in KCPC after pressure shift to higher scattering vector q compared to those of the pristine samples (Fig. R1), revealing a decreased d -spacing from the pressure effect. Besides, the sharper and stronger peaks in the KCPC after pressure indicate a higher crystallinity due to the pressure effect. The peak shift and the similar profile pattern suggest the same crystal symmetry (tetragonal, P4/mmm) in KCPC before and after pressure. Second, Raman spectroscopy measurements indicate that the in-plane bending of carbon-hydrogen bond (δ_{CH}) and the stretching of pyrazine ring (ν'_{ring}) shift to higher frequencies after pressure (Fig. R2), as shown in the revision. The hardened vibrational modes could be attributed by the shorter C-H bond length and smaller pyrazine ring in the compressed KCPC lattice, which is consistent to the WAXS peak shift. Besides, first-principles simulations reveal an enhanced magnetic moment of Cr-pyz layer due to the decrease in magnetic moment of pyrazine molecule when c-axis is compressed by a ratio up to 12% (Fig. R3a-b), while the Cr moment keeps unchanged. The

persisting stability of the initial crystal symmetry is confirmed after the optimization of compressed crystal structure. By pressure, the bond length of Cr-N slightly decreases, indicating the intramolecular deformation of pyrazine (Fig. R3c). The shortened distance of Cr-Cl maintains long enough to avoid a potential chemical reaction (Fig. R3d). The structural transition did not occur even under 12% compression ratio of simulation. In our experiments, the hydrostatic pressure is below 1 GPa, corresponding to a compression ratio of $\sim 2.2\%$ ¹. In metal-pyrazine coordinated molecular magnets,^{1, 2, 3, 4, 5} the pressure induced structural transition is usually at much higher pressure, i.e. 6 GPa. Especially, the structural changes below 1 GPa often come from the modifications of bond length and angle without the symmetry change^{2, 3}. Therefore, according to our additional experimental and simulation results, the lattice change is suggested to induce the pressure effect on KCPC.

Reviewer #3:

The authors have well addressed my questions! I recommend it public in the journal!

Reply: We thank the reviewer for recommendation.

References

1. Halder, G.J., Chapman, K.W., Schlueter, J.A. & Manson, J.L. Pressure-induced sequential orbital reorientation in a magnetic framework material. *Angew Chem Int Ed Engl* **50**, 419-421 (2011).
2. Ghannadzadeh, S. *et al.* Evolution of magnetic interactions in a pressure-induced Jahn-Teller driven magnetic dimensionality switch. *Phys. Rev. B* **87** (2013).
3. O'Neal, K.R. *et al.* Pressure-induced structural transition in copper pyrazine dinitrate and implications for quantum magnetism. *Phys. Rev. B* **93**, 104409 (2016).
4. O'Neal, K.R. *et al.* Spin-Lattice Coupling in [Ni(HF(2))(pyrazine)(2)]SbF(6) Involving the HF(2)(-) Superexchange Pathway. *Inorg Chem* **55**, 12172-12178 (2016).
5. Wehinger, B. *et al.* Giant Pressure Dependence and Dimensionality Switching in a Metal-Organic Quantum Antiferromagnet. *Phys. Rev. Lett.* **121**, 117201 (2018).

REVIEWERS' COMMENTS

Reviewer #1 (Remarks to the Author):

The additional data and calculations have sufficiently answered my questions and improved the main arguments of the paper. I can now support publication of the paper.

Reviewer #2 (Remarks to the Author):

The additional experiments and calculations have addressed my main concern. The revised manuscript has been well improved. I would recommend the publication.

Reviewer #1:

The additional data and calculations have sufficiently answered my questions and improved the main arguments of the paper. I can now support publication of the paper.

Reply: We thank the reviewer's support for publication of our revised manuscript. Those meaningful and helpful comments from the reviewer helped to improve this manuscript for publication.

Reviewer #2:

The additional experiments and calculations have addressed my main concern. The revised manuscript has been well improved. I would recommend the publication.

Reply: We thank the reviewer's recommendation and comments in the review procedure that allow our revised manuscript to be published.